# Optimization and Characterization of PEG Extraction Process for Tartary Buckwheat-Derived Nanoparticles

**DOI:** 10.3390/foods13162624

**Published:** 2024-08-21

**Authors:** Jiyue Zhang, Chuang Zhou, Maoling Tan, Yanan Cao, Yuanhang Ren, Lianxin Peng

**Affiliations:** Key Laboratory of Coarse Cereal Processing of Ministry of Agriculture and Rural Affairs, Chengdu University, Chengdu 610106, China; 17623161272@163.com (J.Z.); zhouchuang@cdu.edu.cn (C.Z.); tanmaoling@cdu.edu.cn (M.T.); caoyanan@cdu.edu.cn (Y.C.); renyuanhang@cdu.edu.cn (Y.R.)

**Keywords:** Tartary-buckwheat-derived nanoparticles, PEG precipitation, process optimization, characterization, antioxidant activity

## Abstract

Plant-derived edible nanovesicles serve as crucial nanocarriers for targeted delivery of bioactive substances, including miRNAs and phytochemicals, to specific tissues. They have emerged as a significant focus in precision nutrient delivery research. In this study, Tartary-buckwheat-derived nanoparticles (TBDNs) were isolated and purified using a combination of differential centrifugation and PEG precipitation. A response surface test was employed to optimize the extraction process of TBDNs in terms of yield, total phenol and flavonoid content, as well as antioxidant activity. The results demonstrated that TBDNs exhibited the highest yield and activity at a 10% concentration of PEG, pH 5, and centrifugation temperature of 4 °C. Under these conditions, the measured yield of TBDNs was 1.7795 g/kg, with a total phenol content of 178.648 mg/100 g, total flavonoid content of 145.421 mg/100 g, and DPPH-radical-scavenging rate reaching 86.37%. Characterization through a transmission electron microscope and nanoparticle-size-tracking analyzer revealed that TBDNs possessed a teato-type vesicle structure with dispersed vesicle clusters present within them. Furthermore, the extracted TBDNs were found to have an average particle size of 182.8 nm with the main peak observed at 162.8 nm when tested for particle size distribution analysis. These findings provide a novel method for extracting TBDNs while laying the groundwork for future investigations into their activities.

## 1. Introduction

Tartary buckwheat (*Fagopyrum tartaryum*) is an edible and medicinal pseudocereal, abundant in diverse nutrients, which demonstrates regulatory effects on gut microbiota and hypoglycemic properties, among other physiological functions. However, the analysis of buckwheat’s mechanism of action based on traditional nutritional functional components remains controversial, suggesting the existence of yet undiscovered bioactive compounds [1]. Plant-derived exosome-like nanovesicles (PELNs) are vesicles with a nano-like structure, containing lipids, proteins, miRNAs, etc. [2]. Numerous studies have demonstrated that exosome-like vesicles (PELNs) of plant origin have a variety of physiological functions that can affect human physiological health, such as regulating gut microbiota, maintaining gastrointestinal homeostasis, and improving enteritis [3]. It has been found that buckwheat-derived nanocapsules (TBDNs) can be absorbed by the gut microbiota; the results of bioconfidence analyses and in vitro validation indicate that TBDNs can regulate the growth of *Lactobacillus rhamnosus* and *Escherichia coli* [4]. In current research, there is a growing awareness of the potential significance of PELNs in disease. PELNs exhibit resistance to digestion by various enzymes, including pepsin and enteropancreatic enzymes, while maintaining stability within the digestive environment [5,6,7,8]. Moreover, they function as regulators of physiological health. For instance, ginger-derived ELNs (GELNs) have been observed to modulate gut microbial composition and enhance intestinal repair, thereby mitigating acute colitis [5,9]. Another study found that tea-derived ELNs were also effective in preventing and treating inflammatory bowel disease (IBD) in mice [7]. In recent years, extracellular vesicles (EVs) have emerged as a promising drug delivery vehicle due to their exceptional stability and safety profiles, as well as their remarkable ability to traverse biological barriers [10,11]. PELNs possess inherent safety, non-toxicity, and low immunogenicity, rendering them highly advantageous natural products [6,12,13,14]. Recent studies have demonstrated the effective delivery of siRNA drugs to target colonic tissues and the inhibition of CD98 expression for treating patients with ulcerative colitis using nanolipids derived from ginger [12].

In recent years, PELNs have been extensively studied for their potential applications in the diagnosis and treatment of various diseases, and there have been a large number of studies on the effects of PELNs on human diseases [15]. Given their current development as alternative therapies for a wide range of diseases, it is imperative to identify an isolation and purification method that is cost-effective and universally compatible in all aspects. PELNs have primarily been isolated through ultracentrifugation, and our research team has also successfully extracted TBDNs using ultracentrifugation in previous studies [4]. The extraction process of PELNs, however, is intricate, accompanied by the common challenges of low purity and yield. Moreover, numerous extraction and separation methods significantly escalate the production cost of PELNs [15]. Therefore, it is necessary to overcome these limitations by improving the extraction process as well as expanding the available sources. Five major classes of methods have been applied to the isolation and purification of these nanoscale vesicles, mainly including ultracentrifugation methods, exosome size-based separation techniques, precipitation techniques, immunoaffinity capture techniques, and microfluidic techniques [15]. Ultracentrifugation and density gradient centrifugation are the best methods for the separation of PELNs, and sucrose gradient centrifugation in particular is keenly sought after by researchers because it provides the theoretically best purity [16]. However, both ultracentrifugation and sucrose gradient density centrifugation have different degrees of drawbacks, such as high cost, limited sample throughput, and a complicated and difficult to master procedure. Several alternative purification methods have been developed to replace the use of ultracentrifugation, such as polymer precipitation and ultrafiltration [17,18,19]. Polyethylene glycol 6000 (PEG6000) precipitation has been successfully used for the purification of mammalian exosomes and viruses. Mechanistically, PEG is known to be a macromolecular crowding reagent and is a highly reticulated polymer where some nanovesicles can be trapped and precipitated from it [20]. Since animal and plant-like exosomes are similar in structural features, the present study utilized a cost-effective polyethylene glycol (PEG6000) precipitation method for the isolation and purification of TBDNs. The effects of different pH, PEG concentration, and temperature on the yield, total phenols, total flavonoids, and antioxidant activity of TBDNs were also investigated to find an optimal extraction condition for TBDNs.

## 2. Materials and Methods

### 2.1. Materials and Instruments

Tartary buckwheat: Chuanqiao No.1, provided by Key Laboratory of Coarse Cereal Processing of Ministry of Agriculture and Rural Affairs, Chengdu, China; Total Plant Phenol (TP) Content Assay Kit, Solarbio, Beijing, China; methanol; sodium hydroxide; 1,1-diphenyl-2-trinitrophenyl hydrazine (DPPH); aluminum nitrate; sodium nitrite; and anhydrous ethanol were analytically pure, purchased from Chengdu Colony Chemicals Co., Ltd., Chengdu, China.

BM-02 Freezing Centrifuge (Shenzhen Dongmei, Shenzhen, China); NanoSight Nano Particle Size Analyser (Malvern Instruments Ltd., Malvern, UK); Biotek synergy HTX Enzyme Labeler (BioTek Instruments, Inc., Winooski, USA); CP214 Analytical Balance (OHAUS Corporation, Parsippany, NJ, USA); QL-901 Vortex Mixer (Kyrin-Bell, Nantong, China); HT7700 Transmission electron microscope (HITACHI High-Technologies Corporation, Tokyo, Japan) were used.

### 2.2. Experimental Methods

#### 2.2.1. Tartary Buckwheat Pretreatment

Tartary buckwheat seeds that were unaffected by pests were selected and cleaned. Dust and impurities were removed and the seeds were dried in an oven at 50 °C for 8 h. Then, the seeds were crushed, ground, and filtered through a 50-mest sieve for later use.

#### 2.2.2. Extraction Process of TBDNs

The powder of Tartary buckwheat seeds was weighed, mixed with an appropriate amount of water, and incubated overnight. The next day, the buckwheat aqueous solution underwent centrifugation at 4 °C at 1000× *g*, for 10 min, 3000× *g*, for 20 min, and 10,000× *g*, for 30 min. Then, the supernatant underwent a 1 μm filter filtration process and PEG6000 incubation overnight. On the next day, centrifugation was performed at 10,000× *g* for 30 min at 4 °C, and the supernatant was removed to leave a precipitate. Excess supernatant was then removed using a piece of tissue paper. The exosomes were freeze-dried and weighed to calculate their yield.

#### 2.2.3. Optimization of the Extraction Process of TBDNs

(1)One-way experiment

The effects of three factors, pH (1, 3, 5, 7, 9), PEG concentration (8%, 10%, 12%, 15%, 20%), and temperature (4, 8, 12, 16, and 20 °C), on the extraction of TBDNs were investigated using the yield of TBDNs, total phenolics, total flavonoids, and the scavenging rate (%) of the DPPH free radicals as the indicators.

(2)Response surface experimental design

Based on the results of the one-way test, the yield of TBDNs, total phenols, total flavonoids, and DPPH radical scavenging (%) were used as indicators, and three factor levels of pH, PEG concentration, and temperature were set, and Box–Behnken software was used to design a three-factor, three-level response surface test, and the experimental design is shown in Table 1.

#### 2.2.4. Determination of Biological Activity of TBDNs

(1)Determination of total polyphenol content

The total plant phenol (TP) content was determined using a commercial kit (Solarbio, Beijing, China). Gallic acid was used as the standard, and a 5 mg/mL stock solution was diluted to different concentrations with distilled water to create a standard curve. For sample preparation, approximately 0.1 g of TBDNs was extracted with 2.5 mL of methanol using an ultrasonic extraction method (power: 300 W, crushing time: 5 s, intermittent time: 8 s, temperature: 60 °C, duration: 30 min). The extract was then centrifuged at 12,000 rpm and 25 °C for 10 min, and the supernatant was collected for analysis. For the assay, 10 µL of each diluted standard solution or sample extract was mixed with 50 µL of Folin–Ciocalteu reagent and vortexed for 2 min. Then, 50 µL of sodium carbonate and 90 µL of distilled water were added and vortexed for 10 min. The absorbance was measured at 760 nm. The standard curve of gallic acid was plotted, yielding the equation Y = 3.2665x + 0.0281 (R^2^ = 0.9991). The total phenol content of TBDNs was calculated from this standard curve and expressed as mg/100 g.

(2)Determination of total flavonoid content

The total flavonoid content was determined by aluminum salt chromatography, with reference to the method of Gao Yan [21]. The standard curve of rutin was plotted, and the total flavonoid content in the samples was calculated according to the linear regression equation of the rutin standard curve: Y = 0.138x + 0.0028, R^2^ = 0.9989. The total flavonoid content was expressed as milligrams of rutin equivalent per 100 g of extracted and isolated exosomes (mg/100 g).

(3)Measurement of DPPH Free Radical Scavenging

The sample pretreatment of exosome-like nanodomains (TBDNs) of buckwheat origin was the same as above, and then the method of Wang Jun-Ru [22] was referred to with slight modification. First, 1 mL of the aqueous solution of the sample is added to 0.1 mg/mL of DPPH solution and 1 mL of ethanol solution, mixed, and reacted for 30 min at room temperature while avoiding light, then the absorbance was measured at 517 nm and the DPPH-radical-scavenging rate was calculated.

#### 2.2.5. Characterization of TBDNs

(1)Electron microscopic detection of TBDNs

First, 20 µL of the prepared TBDNs sample was dropped onto a copper grid and left for 5–10 min for natural adsorption, and then the excess droplets were blotted dry with filter paper; after drying them, an equal amount of 2% phosphotungstic acid solution was taken out and allowed to stand for 3–5 min for staining, followed by removing the excess droplets and air-drying them under incandescent lamps; finally, photographs were taken under the light of a transmission electron microscope for observation.

(2)Particle size detection of TBDNs

Nanoparticle size and surface potential were measured using a Malvern zeta sizer nano ZS (Malvern Instruments, Malvern, UK). Samples were diluted in PBS and three measurements were taken for each sample at room temperature. Particle sizes of at least three independent batches were measured and the mean ± standard deviation calculated.

#### 2.2.6. Statistics and Analysis of Data

All data obtained from the experiments were statistically analyzed and plotted using GraphPad Prism 9.5.0 software and expressed as mean ± standard deviation, and all experimental results were the result of three independent replications. Design-Expert 10.0 software was used to design the response surface experimental protocol and regression analysis of the experimental results.

## 3. Result and Discussion

### 3.1. Results of the One-Way Test

#### 3.1.1. Effect of Different Extraction Processes on the Extraction Yield of TBDNs

Polyethylene glycol (PEG6000)-based enrichment is a convenient and easily scaled-up method for exosome and virus purification [18,23], and many PELNs have been successfully isolated and extracted using this method [24,25]. However, some studies have found that changes in the concentration of PEG can have an effect [26], while another study found that different pH can have an effect on ginger exosome-like vesicles [27]. In addition, temperature is an important factor that affects the relevant extraction rate as well as biological activity. Therefore, we attempted to test the effect of variations in these factors on the extraction yield of TBDNs. A combination of differential centrifugation and PEG precipitation was used to isolate and extract exosome-like vesicles from buckwheat of edible plant origin. Firstly, the buckwheat powder solution was centrifuged at different temperatures (4 °C, 8 °C, 12 °C, 16 °C, 20 °C), and the supernatant was taken through a series of centrifugation steps and adjusted to different pH levels (1, 3, 5, 7, 9); different concentrations of PEG6000 were added to make the final concentration (8%, 10%, 12%, 15%, 20%), and the solution was incubated overnight at 4 °C and 10,000 g TBDNs were obtained after centrifugation for 30 min, dried, and weighed.

As shown in Figure 1, we noticed that higher yields of TBDNs were obtained at acidic pH compared to neutral and basic pH conditions. Also, we observed that the yield of TBDNs was higher at PEG final concentrations of 12% and 15% and that the yield of TBDNs decreased with increasing temperature. This is similar to some reports, which showed that ginger exosome-like vesicles had higher yields, contained higher levels of phenolics and lipids, and exhibited greater in vitro antioxidant activity by varying the acidity of pH [5,27]. From previous reports, acidic conditions are a suitable environment for the presence and isolation of exosomes, and acidic pH can increase the stability of exosomes in vitro, leading to the isolation of higher yields of exosomes [28,29]. The utilization of a PEG-based isolation technique has been widely acknowledged for its ability to effectively preserve the structural integrity of nanovesicles by entrapping them within a mesh-like net formation [26]. Consequently, employing PEG precipitation at lower pH levels can significantly enhance the yield of PDNPs without compromising their essential bioactive components and overall integrity [27]. However, extremely acidic conditions may lead to degradation or denaturation of the bioactive ingredient, instead reducing the yield.

#### 3.1.2. Effects of Different Extraction Processes on the Total Phenolic Content of TBDNs

As above, from the effect of different factors on the total phenolic content of TBDNs, it can be found (Figure 2) that the total phenolic content of TBDNs was higher in acidic conditions than in alkaline conditions. It has been found that ginger-derived exosome-like nanovesicles (PDNPs) have higher total phenolic content at pH 4 and pH 5 [27], which is similar to our results. This may be related to the fact that phenolic components are more stable under acidic conditions. However, too-low acidity will also promote the degradation of phenols. The total phenol content of the nanoparticles was significantly enhanced when exposed to a final PEG concentration of 10% and maintained at a temperature of 4 °C, suggesting that under these optimized conditions, an ideal network structure can be formed to facilitate efficient entrapment and precipitation of the nanoparticles.

#### 3.1.3. Effects of Different Extraction Processes on the Total Flavonoid Content of TBDNs

From the effects of different factors on the total flavonoid content of TBDNs, it can be found (Figure 3) that the total flavonoid content of TBDNs is higher in acidic conditions than in alkaline conditions; this one-way experiment shows that the total flavonoid content of TBDNs is better at pH 5, PEG final concentration of 10%, and temperature of 4 °C.

#### 3.1.4. Effect of Different Extraction Processes on the Antioxidant Activity of TBDNs

The results of this one-way experiment were similar to the above results, and we found that the DPPH-radical-scavenging rate of TBDNs was stronger in acidic conditions than in alkaline conditions (Figure 4), which may be attributed to the higher content of total flavonoids and total phenols in TBDNs under acidic conditions. This result is also similar to previous studies, where it was found that plant-derived nanoparticles (PDNPs) at pH 4 or pH 5 had more antioxidant activity.

### 3.2. Response Surface Optimization Tests

#### 3.2.1. Response Surface Optimization and ANOVA Results

The analysis of variance and significance of the model was performed on the data in Table 2 using Design-Expert 10.0 software, and the regression model equations were obtained as follows: y yield = 1.74 − 0.14 × A + 0.11 × B − 0.091 × C − 0.3 × AB − 0.077 × AC − 0.14 × BC − 0.3 × A2 − 0.2 × B2 − 0.049 × C2; Y DPPH-free-radical-scavenging rate = 86.34 − 3.24 × A − 2.34 × B − 2.71 × C − 2.12 × AB − 0.41 × AC − 2.69 × BC − 11.3 × A2 − 5.42 × B2 − 0.48 × C2; Y total phenolic content = 179.61 − 9.17 × A − 8.79 × B − 8.55 × C + 3.95 × AB − 8.83 × AC + 12.09 × BC − 20.44 × A2 − 14.07 × B2 − 2.15 × C2; Y total flavonoid content = 144.69 − 4.53 × A − 4.08 × B − 2.87 × C − 2.72 × AB + 5.13 × AC + 3.63 × BC − 14.01 × A2 − 12.5 × B2 − 2.84 × C2. The results of ANOVA for the regression model are shown in Table 3, Table 4, Table 5 and Table 6, and from these regression analyses it can be seen that the F-values of the models were all significant (*p* < 0.0001) and out-of-fit terms (*p* > 0.05), indicating a good model fit [30,31]. The corrected coefficients of determination of the model, Adj R^2^, were all greater than 0.9, indicating that the model could explain more than 90% of the variation in the four indicators of TBDNs from pH, PEG concentration, and temperature. Therefore, the extraction process of TBDNs can be better predicted and analyzed with this model. From the F-value, it can be seen that the effects of the three single factors on the yield of TBDNs were in the order of pH (A) > PEG concentration (B) > temperature (C); on the scavenging rate of DPPH radicals they were in the order of pH (A) > temperature (C) > PEG concentration (B); on the content of total phenols they were in the order of pH (A) > PEG concentration (B) > temperature (C); and on the content of total flavonoids they were in the order of pH (A) > PEG concentration (B) > temperature (C). Response surface analysis showed that the effect of pH was the most significant (*p* < 0.01). Therefore, it indicates that pH is an important influencing factor.

#### 3.2.2. Effect of Factor Interactions on TBDNs

The steeper the response surface and the more elliptical the shape of the contour plot, the more significant effect of the two interactions on TBDNs [32,33,34,35]. Based on the results of the regression equation, we further analyzed the shape of the response surface plots and contour plots to determine the effects of pH (A), PEG (B), and temperature (C) on TBDNs. The effects of different factor interactions on TBDNs are shown in Figure 5 and Figure 6. Analysis of the shapes of these response surface plots and contour plots showed that the interactions of the three factors, namely, pH, PEG concentration, and temperature, were significant (*p* < 0.05), and the surface plots as well as contour lines were steep and elliptical. Therefore, the factor interaction was consistent with the ANOVA results. Finally, the optimal process for the extraction of TBDNs was preferred by the analysis and optimization of Design-Expert 10.0 software.

#### 3.2.3. Optimal Process Validation

After Design-Expert 10.0 software analysis and optimization, we finally obtained the optimal parameter process predicted by the TBDN extraction model through the response surface software as follows: pH of 4.736, PEG final concentration of 9.812%, and temperature of 4 °C. The model predicted a yield of 1.75 g/kg of TBDNs, 88.63% DPPH-radical-scavenging rate, 187.586 mg/100 g of total phenols, and 146.332 mg/100 g of total flavonoids at a temperature of 4 °C. For the convenience of the experimental operation, the final process parameters were determined to be pH 5, 10% final concentration of PEG, and 4 °C, and the yield of TBDNs was measured to be 1.7795 g/kg, 86.37% DPPH-radical-scavenging rate, 178.648 mg/100 g of total phenols, and total flavonoid content of 145.421 mg/100 g. The close similarity between the predicted and experimental values demonstrates the stability and reliability of the optimized parameters derived from the model. This strong correlation suggests that the model effectively captures the key factors influencing TBDN extraction.

It is important to acknowledge potential limitations of our study. The model focuses on three key parameters (pH, PEG concentration, and temperature), but other factors such as extraction time and solvent-to-solid ratio also influence the extraction efficiency. Future studies could incorporate these additional variables to further refine the extraction process.

### 3.3. Characterization of TBDNs

Currently, the combination of differential centrifugation and PEG precipitation is used for the isolation and purification of PELNs from foods such as ginger, blueberries, garlic, yam, and tomatoes [9,36,37,38]. The identification of exosome-like nanovesicles derived from plants often involves a combination of techniques to supplement their characterization, particularly in terms of their visual appearance, shape, and dimensions. Therefore, in practical applications, after determining the size and concentration of these nanovesicles using NTA analysis, individual vesicles are commonly examined using scanning electron microscopy (SEM) and atomic force microscopy (AFM) [16]. In this study, a sample of TBDNs was isolated from buckwheat by a combination of differential centrifugation and PEG precipitation. Similar to our previous study [4], a typical teato-type vesicle structure was able to be observed under a transmission electron microscope as shown in Figure 7A. The sample was analyzed using a nanoparticle-size-tracking analyzer, revealing the presence of a polydisperse cluster of vesicles. The extracted TBDNs underwent particle size testing (Figure 7B), yielding an average particle size of 182.8 nm with the main peak observed at 162.8 nm. The particle size observed in this study is larger compared to our previous findings obtained through ultracentrifugation, where the average particle size was 141.8 nm with the main peak occurring at 114.8 nm [4]. The above data demonstrate the successful extraction of TBDNs from Tartary buckwheat.

## 4. Conclusions

The PEG precipitation method used in this study is a simple and inexpensive extraction method, which is of use in the large-scale extraction and isolation of plant-derived exosomes. Response surface methodology, which can intuitively analyze the optimal conditions for extraction and can take into account the interactions between multiple factors, has been widely used in optimization studies of extraction process conditions [39,40]. For the first time, we derived from response surface optimization that the yield and activity of TBDNs were optimal at 10% concentration of PEG, pH 5, and centrifugation temperature of 4 °C. The measured yield of TBDNs under the optimal process was 1.7795 g/kg, total phenol content was 178.648 mg/100 g, total flavonoid content was 145.421 mg/100 g, and DPPH-radical-scavenging rate was 86.37%. Characterization by transmission electron microscopy and nanoparticle-size-tracking analyzer revealed that the TBDNs had a teato-type vesicle structure and showed many dispersed clusters of vesicles. The extracted TBDNs were tested for particle size, resulting in an average particle size of 182.8 nm, with the main peak of the particle size at 162.8 nm. The results showed that the extraction process could be used for the extraction of TBDNs.

## Figures and Tables

**Figure 1 foods-13-02624-f001:**
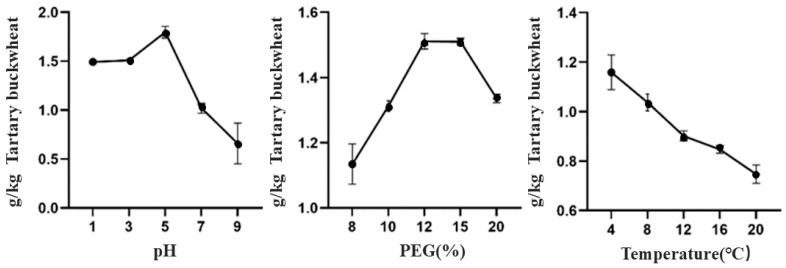
Effect of different factors on the yield of TBDNs.

**Figure 2 foods-13-02624-f002:**
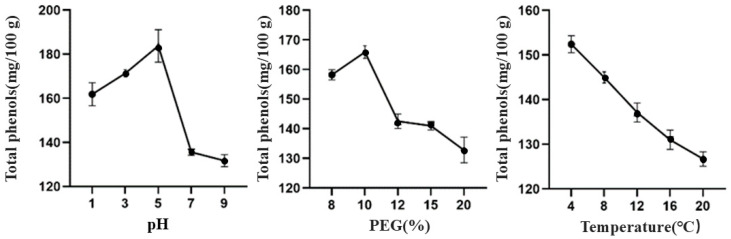
Effect of different factors on the total phenolic content of TBDNs.

**Figure 3 foods-13-02624-f003:**
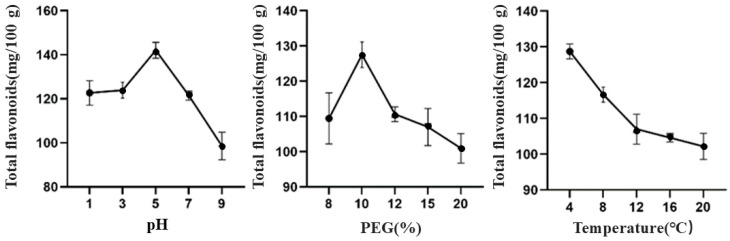
Effect of different factors on the total flavonoid content of TBDNs.

**Figure 4 foods-13-02624-f004:**
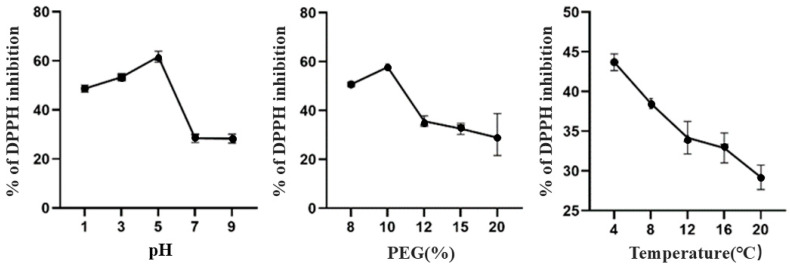
Effect of different factors on the antioxidant activity of TBDNs.

**Figure 5 foods-13-02624-f005:**
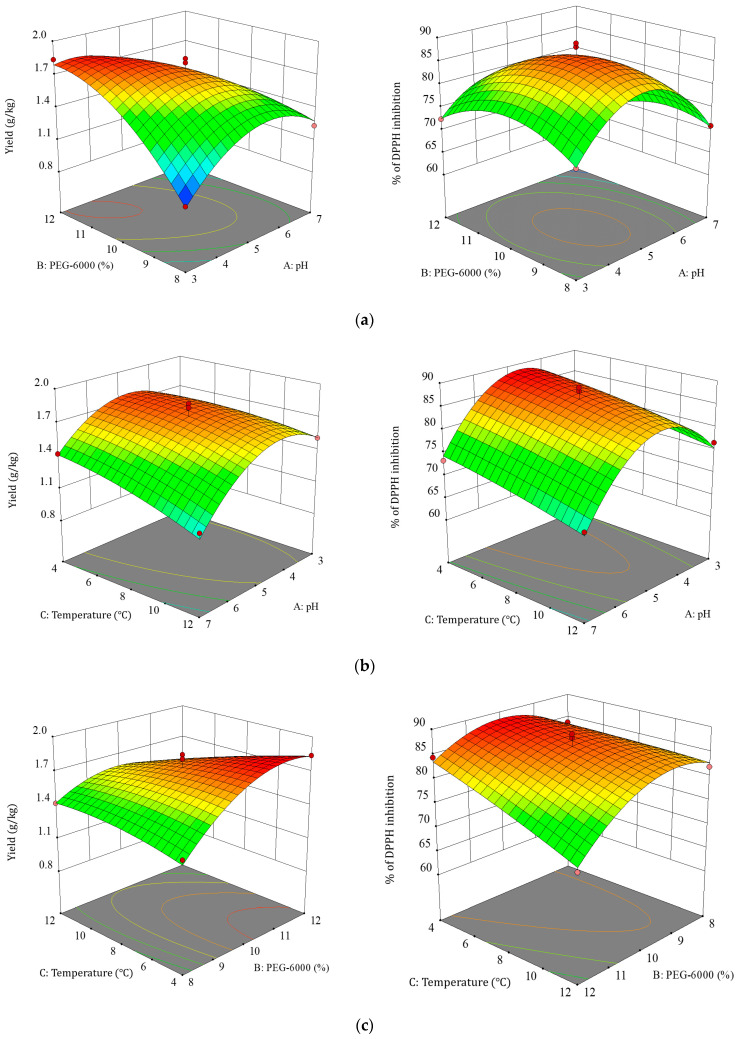
Effect of factor interactions on the yield and antioxidant activity of TBDNs. (**a**) Interaction of PEG concentration and pH. (**b**) Interaction of temperature and pH. (**c**) Interaction of temperature and PEG concentration.

**Figure 6 foods-13-02624-f006:**
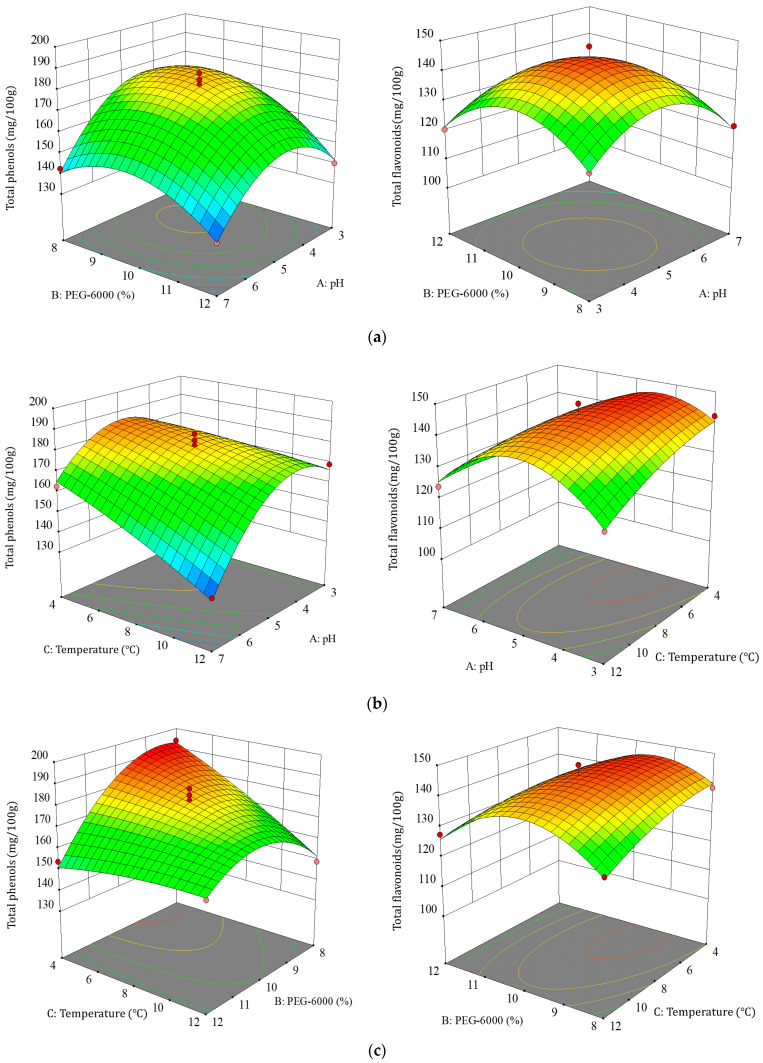
Effect of factor interactions on total phenols and total flavonoids of TBDNs. (**a**) Interaction of PEG concentration and pH. (**b**) Interaction of temperature and pH. (**c**) Interaction of temperature and PEG concentration.

**Figure 7 foods-13-02624-f007:**
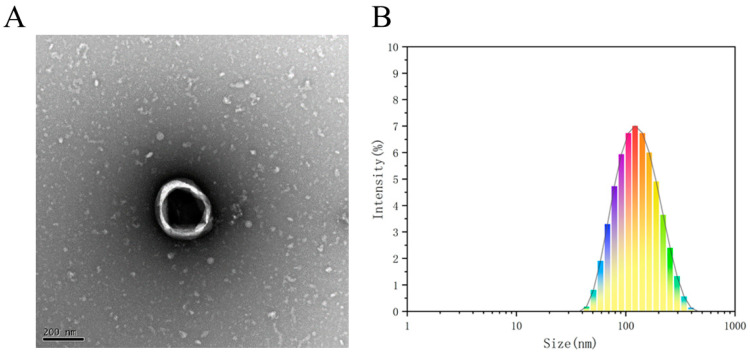
Characterization of TBDNs. (**A**) Electron microscopy diagram of TBDNs. (**B**) Nanoparticle size analyzer measurement of the TBDN particle size.

**Table 1 foods-13-02624-t001:** Experimental Design Table for 3-Factor Response Surface Analysis.

Factor	Level		
	−1	0	1
A: pH	3	5	7
B: PEG6000 (%)	8	10	12
C: temperature (°C)	4	8	12

**Table 2 foods-13-02624-t002:** Response surface experimental design and results.

Number	A	B	C	Y1: Yield (g/kg)	Y2: DPPH Inhibition (%)	Y3: Total Phenol (mg/100 g)	Y4: Total Flavonoid (mg/100 g)
1	5	8	12	1.419	81.900	148.85	127.54
2	5	12	12	1.325	71.77	157.22	127.54
3	3	8	8	0.973	72.85	167.12	123.91
4	5	10	8	1.662	84.43	172.94	148.07
5	7	8	8	1.235	71.00	142.43	121.50
6	3	12	8	1.836	72.46	139.87	120.29
7	5	10	8	1.845	88.00	182.94	143.24
8	7	12	8	0.901	62.15	131.00	107.00
9	5	10	8	1.808	88.77	180.49	148.07
10	5	10	8	1.731	85.75	186	142.03
11	3	10	4	1.495	79.40	164.88	142.03
12	7	10	12	1.129	68.88	131.51	123.91
13	7	10	4	1.42	73.34	162.63	121.50
14	3	10	12	1.51	76.59	169.06	123.91
15	5	12	4	1.838	84.35	153.75	123.91
16	5	10	8	1.631	84.74	175.69	142.03
17	5	8	4	1.355	83.71	193.75	138.41

**Table 3 foods-13-02624-t003:** Results of regression analysis of yield model and regression coefficients.

Source	Sum of Squared Deviations	Degree of Freedom	Mean Square	F-Value	*p*-Value	Significance
Model	933.18	9	103.69	33.16	<0.0001	**
A: pH	84.05	1	84.05	26.88	0.0013	**
B: PEG6000	43.85	1	43.85	14.03	0.0072	**
C: temperature	58.64	1	58.64	18.76	0.0034	**
AB	17.89	1	17.89	5.72	0.048	*
AC	0.68	1	0.68	0.22	0.655	
BC	29	1	29	9.28	0.0187	*
A2	537.78	1	537.78	172.01	<0.0001	**
B2	123.76	1	123.76	39.58	0.0004	**
C2	0.99	1	0.99	0.32	0.5919	
Residual	21.89	7	3.13			
Misfit term	6.67	3	2.22	0.58	0.6564	ns
Pure error	15.22	4	3.8			
Total	955.07	16				
R^2^ = 0.9771 Adj R^2^ = 0.9476 Pre R^2^ = 0.8634						

Note: *p* < 0.01 is highly significant and is denoted by **, *p* < 0.05 is significant and is denoted by *, *p* > 0.05 is non-significant and is denoted by ns.

**Table 4 foods-13-02624-t004:** Results of regression analysis of DPPH-free-radical-scavenging rate model and regression coefficients.

Source	Sum of Squared Deviations	Degree of Freedom	Mean Square	F-Value	*p*-Value	Significance
Model	5631.38	9	625.71	29.22	<0.0001	**
A: pH	672.71	1	672.71	31.42	0.0008	**
B: PEG6000	617.94	1	617.94	28.86	0.001	**
C: temperature	584.31	1	584.31	27.29	0.0012	*
AB	62.57	1	62.57	2.92	0.1311	**
AC	311.52	1	311.52	14.55	0.0066	*
BC	584.91	1	584.91	27.32	0.0012	
A2	1759.09	1	1759.09	82.16	<0.0001	**
B2	833.21	1	833.21	38.92	0.0004	**
C2	19.5	1	19.5	0.91	0.3717	**
Residual	149.88	7	21.41			
Misfit term	37.33	3	12.44	0.44	0.7357	ns
Pure error	112.55	4	28.14			
Total	5781.26	16				
R^2^ = 0.9741 Adj R^2^ = 0.9407 Pre R^2^ = 0.8663						

Note: *p* < 0.01 is highly significant and is denoted by **, *p* < 0.05 is significant and is denoted by *, *p* > 0.05 is non-significant and is denoted by ns.

**Table 5 foods-13-02624-t005:** Results of regression analysis of total phenol content model and regression coefficients.

Source	Sum of Squared Deviations	Degree of Freedom	Mean Square	F-Value	*p*-Value	Significance
Model	5631.38	9	625.71	29.22	0.0001	**
A: pH	672.71	1	672.71	31.42	0.0008	**
B: PEG6000	617.94	1	617.94	28.86	0.001	**
C: temperature	584.31	1	584.31	27.29	0.0012	*
AB	62.57	1	62.57	2.92	0.1311	**
AC	311.52	1	311.52	14.55	0.0066	*
BC	584.91	1	584.91	27.32	0.0012	
A2	1759.09	1	1759.09	82.16	<0.0001	**
B2	833.21	1	833.21	38.92	0.0004	**
C2	19.5	1	19.5	0.91	0.3717	**
Residual	149.88	7	21.41			
Misfit term	37.33	3	12.44	0.44	0.7357	ns
Pure error	112.55	4	28.14			
Total	5781.26	16				
R^2^ = 0.9741 Adj R^2^ = 0.9407 Pre R^2^ = 0.8663						

Note: *p* < 0.01 is highly significant and is denoted by **, *p* < 0.05 is significant and is denoted by *, *p* > 0.05 is non-significant and is denoted by ns.

**Table 6 foods-13-02624-t006:** Results of regression analysis of total flavonoid content model and regression coefficients.

Source	Sum of Squared Deviations	Degree of Freedom	Mean Square	F-Value	*p*-Value	Significance
Model	2195.81	9	243.98	32.45	<0.0001	**
A: pH	164.08	1	164.08	21.83	0.0023	**
B: PEG6000	133.01	1	133.01	17.69	0.004	**
C: temperature	65.84	1	65.84	8.76	0.0211	*
AB	29.59	1	29.59	3.94	0.0877	**
AC	105.37	1	105.37	14.02	0.0072	*
BC	52.56	1	52.56	6.99	0.0332	
A2	826.77	1	826.77	109.98	<0.0001	**
B2	657.92	1	657.92	87.52	<0.0001	**
C2	33.91	1	33.91	4.51	0.0713	**
Residual	52.62	7	7.52			
Misfit term	13.52	3	4.51	0.46	0.7246	ns
Pure error	39.1	4	9.78			
Total	2248.44	16				
R^2^ = 0.9766 Adj R^2^ = 0.9465 Pre R^2^ = 0.8766						

Note: *p* < 0.01 is highly significant and is denoted by **, *p* < 0.05 is significant and is denoted by *, *p* > 0.05 is non-significant and is denoted by ns.

## Data Availability

The original contributions presented in the study are included in the article, further inquiries can be directed to the corresponding author.

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
