# Peer review of "Optimization and Characterization of PEG Extraction Process for Tartary Buckwheat-Derived Nanoparticles"

_foods, 2024, doi:10.3390/foods13162624_

Round 1
Reviewer 1 Report
Comments and Suggestions for Authors
The manuscript submitted by the authors presents interesting study of bioactive compounds and antioxidant activities Tartary buckwheat. The topic and study are interesting. The methodology is described correctly with minor comments. However, there is no discussion of the results obtained and justification of the established parameters of the extraction process.
The work is worth publishing, however, authors should add a discussion and take into account the following comments:
Keywords need to be changed to ones more relevant to the topic of the article.
Line 128 Is the equation written correct?
Line 133 What solvent was used for extraction? What were the authors' reasons for using this type of solvent?
Line 140 The results are expressed as mg/ per gram, should be expressed per gram of dry matter.
Line 147 Was it 100 g of sample or 100 g of dry sample?
Line 147 -148 What does this sentence mean: “The total flavonoid content was expressed as the rutin equivalent per 100 g of sample, and the unit was mg/100g.”
Line 284 What is supplement 6?
Result and Disscussion
How is the yield of TBDN was determined?
Please try to explain why the yield of TBDNs, the total phenolic content of TBDNs, the total flavonoid content of TBDNs and the antioxidant activity of TBDNs was the best for pH 5. The authors stated that, in general, better values were obtained in an acidic environment, but it is growth from pH 1 to 5. Why?
Similarly, please explain for the PEG concentration the occurrence of a maximum at 10%.
3.2.3. Optimal process validation
The authors did not conduct any discussion.
3.3. Characterisation of TBDNs
Please describe in detail what teato vesicles are and how they were diagnosed. How does this affect the characteristics of the TBDNs. The results should be compared with literature data.
Please show the cluster of vesicles in the photo. It is difficult to characterize the system based on one vesicles in the photo. Please show many dispersed clusters of vesicles.
Tables 2 to 6 check Significance
Reviewer 2 Report
Comments and Suggestions for Authors
The presented research, titled: Optimization and characterization of PEG extraction process 1 for Tartary buckwheat -derived nanoparticles, outlines the extraction method of exosome-like vesicles. However, the title is not adequate to the presented research. The authors do not confirm the obtaining of TBDNs (tracted buckwheat-derived exosome-like vesicles) in their study. They do not present a detailed characterization of the obtained seed fractions. The methodologies presented are not properly described in a way that allows repetition of the experiment. The authors do not provide information on the number of repetitions. The authors have not accurately considered the research problem in context to studies published in the recent literature. In its current form, the manuscript is not suitable for publication in such a reputable journal.
Comments on the Quality of English LanguageMinor editing of English language required
